

# Terradrönare
## Adaptive control of UAV in territory exploration



**Authors**: Andrzej Krzyśków⊙ · Kamil Taras⊙ · Szymon Maksymowicz⊙ · Zuzanna Słapek⊙

**Supervisor:** Krystian Wojtkiewicz, Department of Applied Informatics, Faculty of Information and Communication Technology

### Abstract

Terradrönare is a solution that aims to deter wild boars from damaging crops by using unmanned aerial vehicles (UAVs) equipped with manually operated scent dispensers. The dispenser releases balls filled with wolf scent, an effective natural deterrent for boars. The system incorporates image recognition technology to help users identify optimal target sites, such as suitable tree trunks, to deploy the deterrents. By reducing direct human interaction with wildlife, Terradrönare improves people's and animals' safety while minimizing the need for manual crop monitoring. Acknowledging potential legal restrictions on semi-automated flight, the solution avoids autonomous operations and provides real-time guidance through a mobile Android application displayed on the user's device. This ensures legal compliance while effectively managing wildlife and protecting agricultural resources.

## INTRODUCTION

Efficient environmental monitoring and wildlife management remain a pressing challenge, particularly in addressing the impact of wild boars on forest ecosystems. This project aims to develop an adaptive UAV system capable of territory exploration, tree recognition, wild boar damage detection, and scent distribution for animal control. Integrating advanced technologies will enhance monitoring efficiency, reduce human intervention, and ensure wildlife safety through non-invasive management techniques.

The UAV system introduces an innovative approach by employing a wolf-scent pheromone dispenser to deter wild boars from vulnerable areas, providing an eco-friendly alternative to conventional methods such as fencing or labor-intensive processes such as manual repellents. Additionally, the UAV's capability to recognize environmental features and assess damage in real time enhances monitoring precision and supports operators in making informed decisions about wildlife management and habitat protection.

The project aims to develop a fully functional, field-tested UAV platform compliant with local regulations, featuring a mobile app for control, image recognition algorithms for environmental analysis, and hardware components, including a scent distribution mechanism for scent distribution. By employing advanced recognition technology, the system minimizes time and labor requirements while enhancing the accuracy and efficiency of forest monitoring tasks, offering a practical solution for wildlife management and habitat protection.

### Related Work

Modern agricultural drones, such as *DJI's Agras* series [1] and *Auto Spray Systems* [2], are designed for precision tasks like delivering pesticides and fertilizers while reducing waste. These systems incorporate advanced imaging for accurate target identification and autonomous navigation for efficient coverage, enabling precise delivery to specific areas. Similarly, platforms like *Croptracker* use multi-spectral imaging to monitor crop health, allowing for targeted analysis of features such as tree canopies [3]. Technologies such as automated flight paths and precision spraying demonstrate their adaptability for wildlife management tasks involving similar requirements.

In wildlife applications, drones are primarily employed for passive observation. *Wildlife Drones* integrates radio receivers with UAVs to track animals tagged with VHF radio transmitters, enabling real-time location monitoring over large areas [4]. The *JOUAV CW-15* drone specializes in long-range habitat surveys, while the *JOUAV CW-25 VTOL* drone combines thermal imaging and GPS for tracking endangered species like Māui dolphins [5]. The *DJI Matrice 300 RTK* drone excels in short-range observations such as monitoring nesting birds or responding to environmental disasters [6]. Programs such as Air Shepherd leverage AI for species identification, behavioral analysis, and anti-poaching operations, highlighting how UAVs are increasingly tailored to ecological challenges [7].

## PROPOSED SOLUTION

The development of the adaptive UAV system encompassed three main components: the Android application for control and user interface, the scent distribution hardware, and the image recognition module for environmental feature detection. Each component underwent iterative development and testing to ensure compatibility, compliance with project requirements, and effective integration into the overall system. This section outlines the design process, challenges encountered, and results achieved for each component.

**Architecture Diagram**

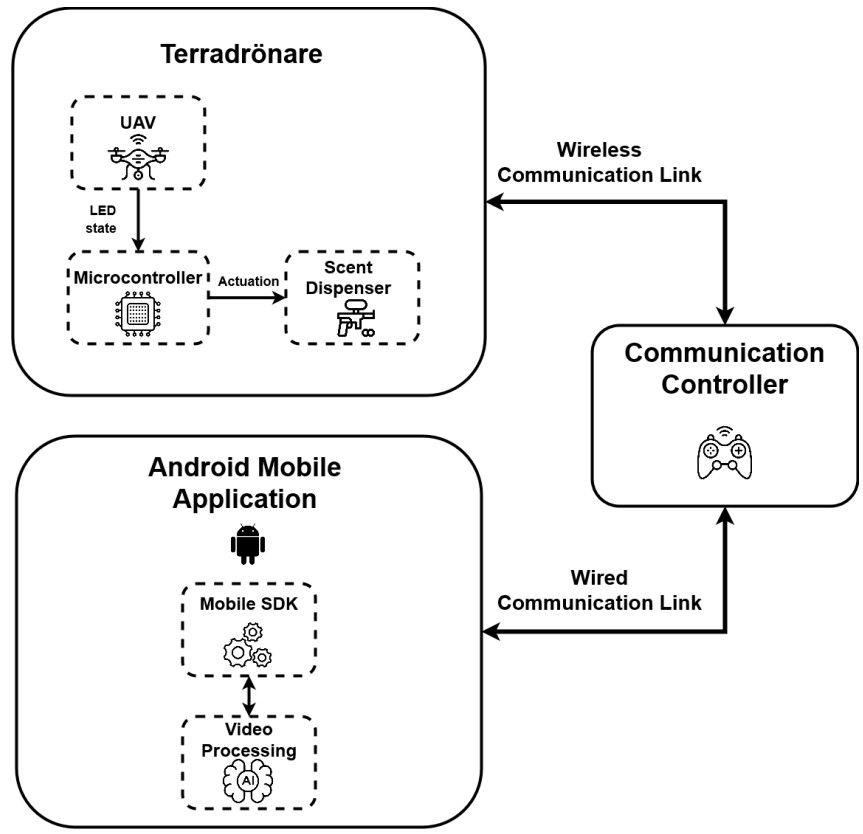

Figure 1: Architecture Diagram [8] [9]

The Terradrönare system operates through coordinated interaction between the user, the Android Mobile Application, and the UAV. The user interfaces with the Android app, which connects to the Communication Controller via a USB-C link. The Communication Controller establishes a wireless OcuSync 2.0 link with the UAV, enabling real-time data transmission and control. Commands from the user are relayed through the app to the UAV, which processes them and signals the Microcontroller. The Microcontroller, using the UAV's LED system, activates the Scent Dispenser to release repellents onto the target. The workflow includes safety measures, such as ensuring the UAV is within range and the "Aim" mode is active before allowing the dispenser to trigger. This ensures precise operation.

**Terradrönare**   The Terradrönare is a UAV integrated with a Scent Dispenser for precise projectile delivery. This design enables synchronized operation and is suitable for environmental and agricultural applications, ensuring accurate targeting and reliable functionality.

**Microcontroller**   The Microcontroller is an Arduino-based system that manages communication between the UAV and the Scent Dispenser. It ensures synchronized operations and accurate firing control, facilitating efficient system coordination.

**Scent Dispenser**   The scent dispenser is a custom module designed to distribute repellents to specific targets, such as trees.

**Wireless Communication Link**   The wireless communication link utilizes OcuSync 2.0, offering dual-band (2.4/5.8 GHz) transmission with a 10-km range, 1080p / 30fps video streaming, and 120 ms latency. It includes anti-interference and multi-device compatibility for robust communication.

**Communication Controller**   The communication controller is based on a DJI system, providing a 10 km range using OcuSync 2.0. It features low-latency video transmission and wired connectivity to mobile devices for precise operation.

**Wired Communication Link**   The wired communication link uses a USB-C connection between the DJI RC 2 controller and a mobile device. This setup ensures stable communication and minimizes latency.

**Android Mobile Application**   The Android application, developed with the DJI Mobile SDK in Java, serves as the central interface for UAV operation. Provides real-time telemetry, live video feeds, and control over the UAV and payload, ensuring communication and firmware validation during the registration process.

**Mobile SDK**   The DJI Mobile SDK supports the integration of drone controls, camera management, telemetry, and video feeds into the Android application. Its support for Java and Kotlin facilitates robust and extensible development.

**Video Processing**   The video processing system uses a YOLO-based machine learning model to detect and classify trees. This capability allows for accurate target tracking in real time, improving operational precision.

## Android Application

The Android application serves as the primary user interface and control system for the UAV platform. Initially, the team explored utilizing a Windows-based software development kit (SDK) for application development. However, compatibility issues with the DJI Windows SDK, including its lack of support, a transition to the DJI Mobile SDK was required. This shift involved redesigning the application framework to accommodate the Android platform.

The Terradrönare application was developed using the DJI Mobile SDK [10] and written in Java. The user interface was designed with a focus on simplicity to facilitate navigation in field conditions. The app provides users with real-time drone telemetry, a live video feed, and control of both the UAV and the scent dispenser, delivering key functionalities essential for UAV operation.

The application includes an initial registration process (Figure 2), requiring users to connect the UAV before accessing the main interface. This step ensures secure communication between the app and the UAV, validating the firmware version and checking compatibility with the DJI SDK for proper operation.

The main screen (Figure 3) of the application displays a live camera feed from the UAV, along with essential real-time telemetry such as altitude, speed, and the distance to the nearest obstacle. These metrics are drawn from the UAV's front cameras, helping operators maintain spatial awareness and avoid collisions. A horizon line overlay further assists in maintaining proper orientation during flight. Two primary control functions are available: the "Aim" function, which activates a crosshair overlay employing the Circular Error Probable (CEP) technique. The crosshair dynamically adjusts its size based on distance measurements from the UAV's front cameras (Figure 4), shrinking as obstacles come closer to indicate increased precision. The "Fire" function triggers the scent distribution mechanism, providing users with control over the UAV's scent deployment system.

The "Fire" function triggers the scent distribution mechanism by activating the UAV's LED lights for a specified period (Figure 5). These lights serve as a signal for the scent mechanism to initiate its sequence. The "Fire" button remains inactive if the "Aim" mode is off or if the UAV is outside the required threshold distance to the target. This ensures that the scent distribution sequence is initiated only under precise conditions, preventing unintended activations and enhancing operational accuracy.

## Scent Dispenser

The final design of the scent dispenser focuses on modularity and simplicity, ensuring a rapid prototyping process. By designing each part of the dispenser to be easily replaceable, we significantly reduced the need to reprint entire components when iterations or changes were required during development. This modular approach, as illustrated in the CAD model of the dispenser (Figure 6), allowed for targeted testing and refinement of specific components, reducing material waste and accelerating the development time.

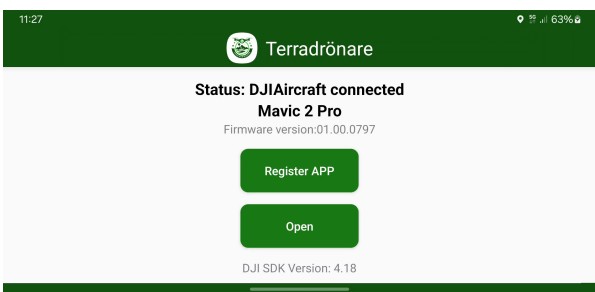

Figure 2: Registering Screen

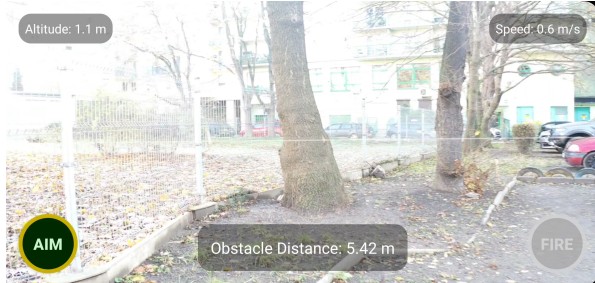

Figure 3: Main Video Screen - Aim Mode Off

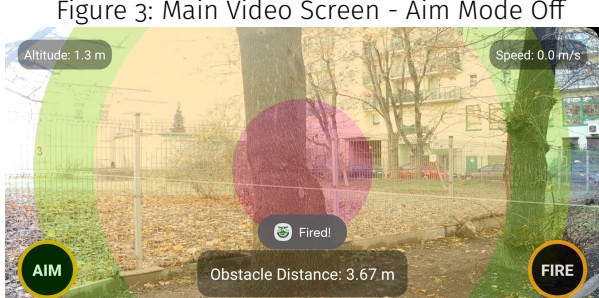

Figure 4: Main Video Screen - Close Up

Figure 5: Scent Dispenser Triggered

A key decision in the development was the selection of a roller-based ball launching mechanism. This mechanism uses two rotating rollers to propel the scent-filled balls forward. This approach was chosen over alternative designs like spring-loaded mechanisms, for its simplicity and reliability. It offers sufficiently consistent performance with fewer moving parts compared to the alternative solutions, reducing the likelihood of mechanical failure and simplifying maintenance.

The dispenser was also designed to seamlessly integrate with the UAV platform, by using the drone's existing components for operation. Specifically, the system uses the drone's onboard LED as a trigger for the launching sequence via a photo-resistor, as detailed in the electrical schematic (Figure 8). This solution was chosen because it eliminated the need for additional communication channels, simplifying the overall system and reducing weight and complexity.

The attachment system securely fastens to the UAV without requiring any modifications to the drone. The 3D-printed version of the dispenser, shown in Figure 7, weighs just under 240g and demonstrates the physical realization of this design, validating its effectiveness and compatibility with the UAV.

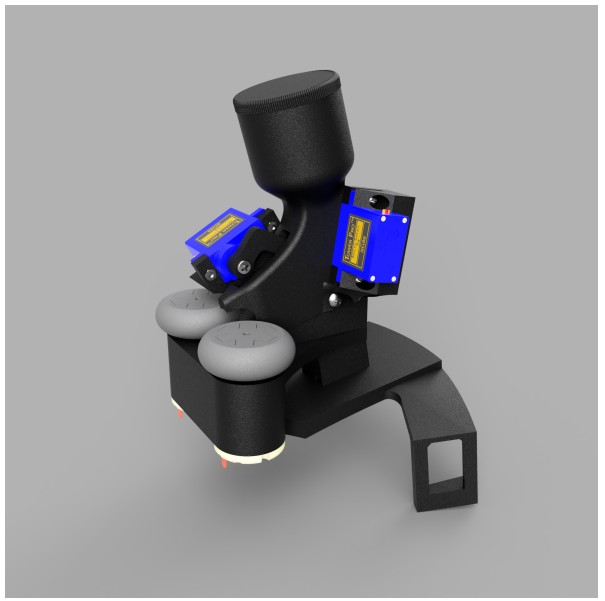

Figure 6: CAD model of the dispenser

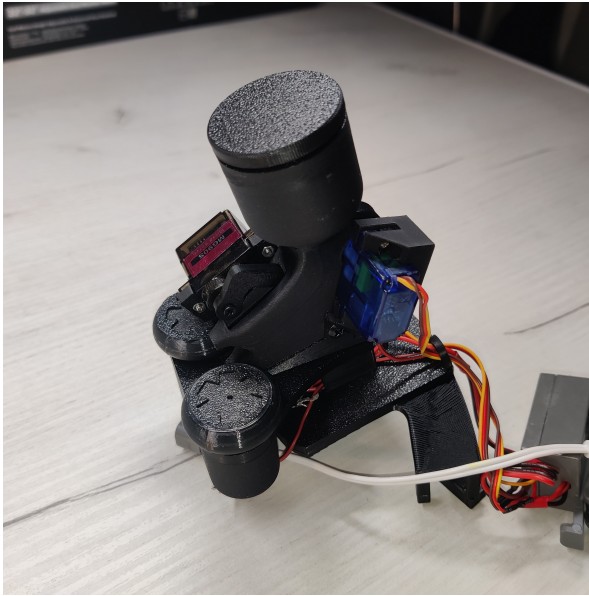

Figure 7: 3D-printed version of the dispenser

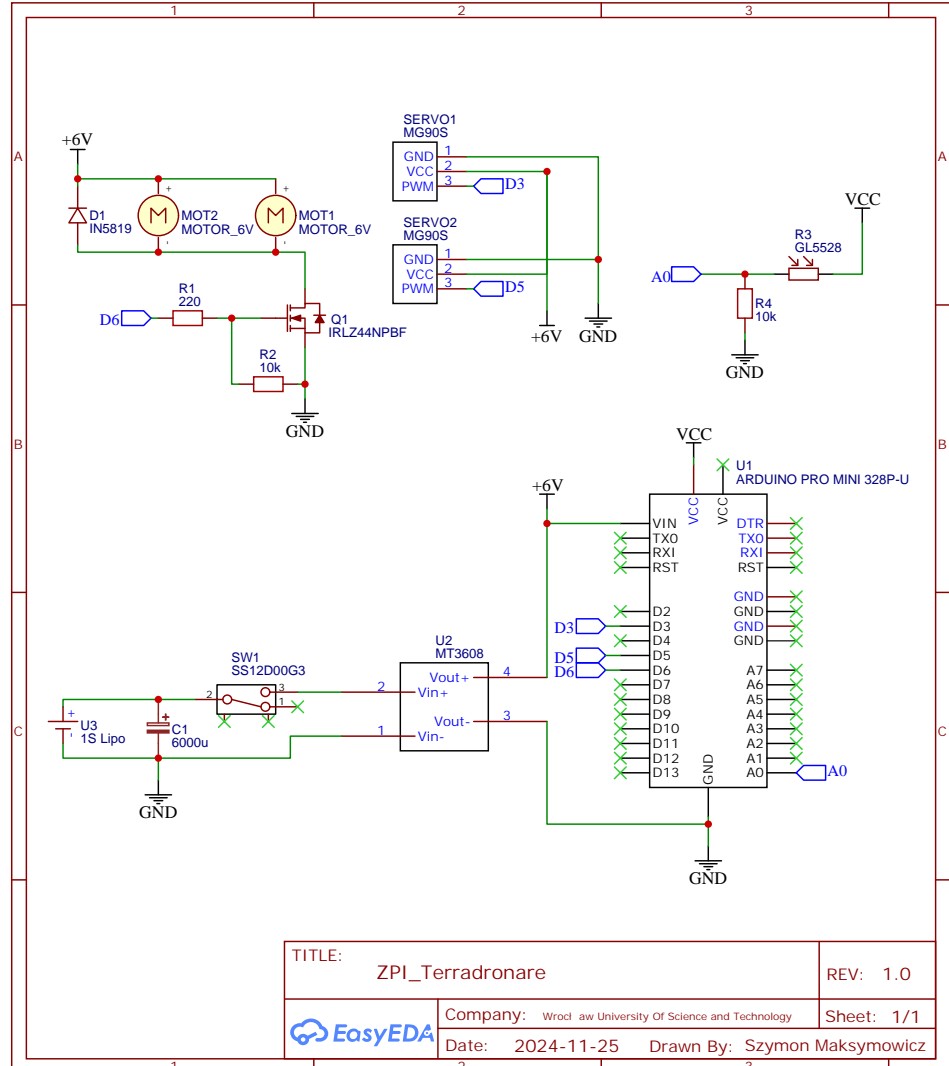

Figure 8: Electrical Schematic

### Image Recognition

The UAV platform integrates object detection and control systems, using a custom-trained *YOLOv8* [11] model and an Android application. The Python-based development focuses on training and evaluating the object detection model, which is then exported to *TensorFlow Lite* [12] for integration into the app. This system combines object detection with system.

The Python part of the workflow focuses on training a YOLOv8 model specifically designed for detecting objects like trees and trunks. Using the Ultralytics library, the model was trained on a custom dataset, achieving high mAP scores across test images. Post-training, the model was evaluated for precision and recall, followed by inference on sample images to validate real-world detection accuracy. The model was then exported to TensorFlow Lite format, optimized for real-time deployment. This resulted in lightweight performance suitable for UAV's live video processing system.

In the Android application, detection is implemented using the TensorFlow Lite model and *OpenCV* [13], integrated with the UAV's live video feed. Frames captured from the UAV camera are processed and passed to the model for real-time inference. The app dynamically visualizes detected objects by drawing bounding boxes and labels directly on the video feed, updating detection results for each frame. The system handles model inference, ensuring detection without lag, even in field conditions, and provides feedback to the user in real-time.

### Testing

The app was extensively tested in simulated and real-world scenarios to ensure reliability. Integration with the DJI Android SDK enabled seamless communication between the mobile device and the UAV, minimizing latency to provide real-time responsiveness. Feedback from field testing was incorporated

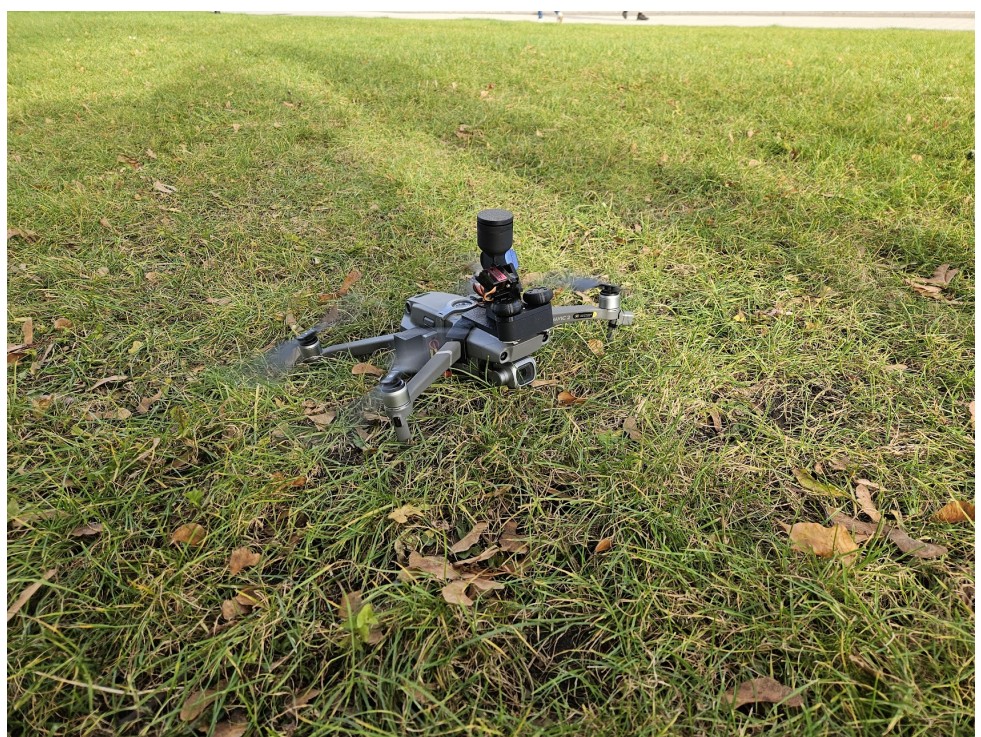

Figure 9: Drone with Scent Dispenser

to refine the interface and improve overall usability. The final application demonstrated stable performance under varying field conditions and successfully supported the intended operational workflows.

## CONCLUSION

This project provided us insights into the complexity and interdisciplinary nature of developing an adaptive UAV system. By integrating electrical components, 3D printing, software engineering, and machine learning, we successfully designed a multifunctional system that connects technical domains. While challenges were inevitable, the progress we achieved underscores the importance of persistence and iterative problem-solving.

The development of the Android application as the control hub was particularly challenging. Initial compatibility issues showed due to the inconsistent support and updates for the Windows SDK, which ultimately necessitated a transition to the DJI Mobile SDK for Android. This highlighted the importance of choosing tools with ongoing support in technical projects. The resulting application delivered real-time telemetry, live video feeds, and precise UAV control. Although there is room for future optimization, the app met the functional requirements needed for reliable operation under field conditions.

The design and implementation of the scent dispenser demonstrated the advantages of a modular approach. Exploiting 3D printing for prototyping enabled rapid iteration and significant savings in time and materials. TPU was chosen as the material for 3D-printed components due to its flexibility, durability, and enhanced friction properties. The increased friction provided by TPU was beneficial for the roller-based mechanism, ensuring grip and movement of components during operation. Despite producing numerous prototypes to perfect the design, the roller-based mechanism was chosen for its simplicity and reliability. Future iterations could focus on enhancing adaptability for broader applications.

The object detection system, utilizing a custom-trained YOLOv8 model, was the most technically demanding component, particularly due to challenges with integrating model into the Android environment. An issue appears from conflicts between OpenCV and Gradle dependencies, where mismatched versions caused build errors. Resolving these conflicts required dependency management, aligning library versions, and customizing Gradle configurations. Regardless of these challenges, the system was optimized for real-time video processing and object detection, demonstrating its suitability for UAV deployment. This underscored the need for careful dependency management to enable future enhancements in object detection accuracy, and performance in complex environments.

Testing the system in both simulated and real-world scenarios pointed up the importance of iterative refinement and user-centered design. However, real-world testing created additional challenges due

to the strict drone laws in Poland, which restricted flight zones and operational conditions. Obtaining the permissions delayed field tests and required modifications to planned testing protocols. These constraints bring out the need for preparation when operating UAVs in regulated environments. Despite these challenges, adjustments ensured the usability under field conditions. Achieving well-grounded performance in tests validated the system's design.

## FUTURE DIRECTIONS

Several potential directions for improvement can be explored, the first being improvements to the battery life of the dispenser attachment and its design. One viable approach is transitioning to a spring-loaded mechanism, where a low-power actuator could release the stored energy quickly on demand. Another promising alternative is utilizing $CO_2$ canisters to drive the dispensing mechanism.

Another area of improvement lies in the integration of a flight plan feature. By allowing the user to define the perimeter of a desired area to be explored, the system can autonomously check the perimeter without much human intervention.

Expanding the image recognition capabilities is also a vital direction for improvement. Currently trained to detect trees, the model could be enhanced to identify more complex features such as signs of wild boar damage or the animals themselves. This would involve training the system on more challenging datasets because such damage is often subtle and varies in appearance depending on the terrain and vegetation. Detecting these patterns accurately may involve integrating more sophisticated sensors, such as thermal imaging or multi-spectral sensors, to provide additional data.

## ACKNOWLEDGMENTS

We want to thank our supervisor, Dr. Krystian Wojtkiewicz, PhD, for providing access to the drone from the university, procuring essential components needed for completing the project and providing constant guidance throughout the project.

We also thank Dr. Piotr Zabawa, PhD, a licensed drone operator, for his contribution to field testing the system, operating the drone for us in an actual usage scenario, and helping us acquire photos of the system in use in the field.

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
