# OpenReview forum: "Terradrönare - Adaptive control of UAV in territory exploration"
_pwr.edu.pl/Wrocław_University_of_Science_and_Technology/2024/ZPI_Day — Wrocław University of Science and Technology 2024 ZPI Day Submission_

### Official Review · Reviewer_L42J · 2024-12-03
**Review of "Terradrönare: Adaptive Control of UAV in Territory Exploration"**

**Confidence:** 4
**Significance Of Results:** 5
**Overall Quality:** 5

**Compliance With Template:**

5: Very High Quality – The article contains all the required sections, which are written in a very detailed, clear, and error-free manner. The structure is professional and meets expectations, and the content adheres to the highest substantive and formal standards.

**Description Of Results:**

4: High Quality – The results are described in detail and supported by usage examples or evaluations. The description is reliable but may lack full depth of analysis.

**Feedback On Consistency:**

The document presents a consistent and logically structured project description. The problem of wild boar management is clearly identified, followed by an innovative UAV-based solution utilizing wolf-scent dispensers. The methodology and results are presented in alignment with the project goals. Each component (e.g., scent dispenser, image recognition module) is described with sufficient technical depth. The conclusions effectively summarize achievements while acknowledging challenges and limitations, such as regulatory restrictions and dependency management. Overall, the narrative maintains a logical flow, connecting the identified problem to the proposed solution and its outcomes.

**Potential For Development:**

The document identifies several opportunities for future development focusing on both hardware and software improvements. For the scent dispenser there is potential to enhance its energy efficiency by transitioning to mechanisms like spring-loaded actuators or CO2 canisters which could provide more reliable and efficient operation. On the software side adding autonomous flight path planning would allow the UAV to operate with minimal human intervention increasing usability in large or complex terrains. Expanding the image recognition capabilities is another promising direction currently trained to detect trees the system could be adapted to identify more complex features such as direct wild boar sightings or subtle environmental damage. These advancements would require training on diverse datasets and potentially integrating additional sensors like thermal or multispectral imaging.

**Project Nature Evaluation:**

Some remarks:
- The solution addresses a pressing issue in wildlife management, offering a practical and eco-friendly alternative to traditional methods like fencing or manual repellents.
- The use of technologies such as YOLOv8 for object detection, real-time telemetry through a mobile app, and modular 3D-printed designs demonstrates advanced technical competence.
- The integration of hardware (scent dispenser, microcontroller) and software (Android application with TensorFlow Lite) reflects a multidisciplinary approach.

However, the document could delve deeper into comparative performance analysis, such as testing against existing wildlife deterrence systems, to strengthen the evaluation of its utility.

**Technical Language Precision:**

5: Very High Quality – The language is entirely appropriate for a technical report. All terms are used correctly and precisely, and the style is professional, clear, and coherent, without any errors or ambiguities.

---

### Official Review · Reviewer_PH9d · 2024-12-05
**Flying canon**

**Confidence:** 5
**Significance Of Results:** 5
**Overall Quality:** 5

**Compliance With Template:**

4: High Quality – The article contains all the required sections, which are well-written and substantively correct, although minor errors or shortcomings may be present. The overall structure is clear and coherent.

**Description Of Results:**

5: Very High Quality – The results are described in detail, clearly and comprehensively, supported by thorough evaluation, analysis, and convincing usage examples. The description meets the highest substantive standards.

**Feedback On Consistency:**

The project is an interesting piece of work. However, the paper doesn't really address the legal issues of flying drones in rural areas. There is also a little evaluation presented.

**Potential For Development:**

There is a lot of potential for development in the project, starting with hardware optimization through AI support strengthening and mobile app design. Nevertheless, even in its current state, the project is mature enough for 10 weeks of students' work.

**Project Nature Evaluation:**

The project described in the paper is a high level of an engineering work..  It comprises both hardware and software aspect with some sophisticated solutions present. The main downside of the presentation is the lack of appropriate results evaluation, e.g., field tests, recognition algorithm metrics, etc.

**Technical Language Precision:**

5: Very High Quality – The language is entirely appropriate for a technical report. All terms are used correctly and precisely, and the style is professional, clear, and coherent, without any errors or ambiguities.

---

### Official Review · Reviewer_ScHM · 2024-12-06
**An innovative project combining several different technologies**

**Confidence:** 3
**Significance Of Results:** 5
**Overall Quality:** 5

**Compliance With Template:**

5: Very High Quality – The article contains all the required sections, which are written in a very detailed, clear, and error-free manner. The structure is professional and meets expectations, and the content adheres to the highest substantive and formal standards.

**Description Of Results:**

5: Very High Quality – The results are described in detail, clearly and comprehensively, supported by thorough evaluation, analysis, and convincing usage examples. The description meets the highest substantive standards.

**Feedback On Consistency:**

The project description is consistent, with a clear analysis of the problem, appropriate use of technical methods and logical presentation of the results.The structure is proper, beginning with identifying the problem, describing the solution and finishing with the results. It's also good that the authors mentioned the problems they encountered throughout the project, especially during the different iterations. As a possible improvement, the team could have provided more details about the test results to better support their findings.

**Potential For Development:**

The team demonstrates a strong potential for development by recognizing opportunities to enhance both the software and hardware aspects of the project. The proposed development ideas are well-chosen and could help improve the project's business potential.

**Project Nature Evaluation:**

The project exhibits strong characteristics of engineering work. It involves the application of several technical methods, including UAVs, mobile technology and 3D design and printing to address a real-world problem.

**Technical Language Precision:**

5: Very High Quality – The language is entirely appropriate for a technical report. All terms are used correctly and precisely, and the style is professional, clear, and coherent, without any errors or ambiguities.

---

### Decision · Program_Chairs · 2024-12-10

Accept (Oral)